# Structural Characterization of Al_65_Cu_20_Fe_15_ Melt-Spun Alloy by X-ray, Neutron Diffraction, High-Resolution Electron Microscopy and Mössbauer Spectroscopy

**DOI:** 10.3390/ma14010054

**Published:** 2020-12-24

**Authors:** Rafał Babilas, Katarzyna Młynarek, Wojciech Łoński, Dariusz Łukowiec, Mariola Kądziołka-Gaweł, Tomasz Czeppe, László Temleitner

**Affiliations:** 1Department of Engineering Materials and Biomaterials, Silesian University of Technology, Konarskiego 18a, 44-100 Gliwice, Poland; rafal.babilas@polsl.pl (R.B.); wojciech.lonski@polsl.pl (W.Ł.); dariusz.lukowiec@polsl.pl (D.Ł.); 2Institute of Physics, University of Silesia, 75 Pułku Piechoty 1, 41-500 Chorzów, Poland; mariola.kadziolka-gawel@us.edu.pl; 3Institute of Metallurgy and Materials Science of Polish Academy of Sciences, 25 Reymonta 5 St., 30-059 Kraków, Poland; t.czeppe@imim.pl; 4Wigner Research Centre for Physics, Konkoly Thege út 29-33, H-1121 Budapest, Hungary; temleitner.laszlo@wigner.hu

**Keywords:** Al-based alloys, X-ray diffraction, neutron diffraction, transmission electron microscopy, differential scanning calorimetry, Mössbauer spectroscopy, electrochemical tests

## Abstract

The aim of the work was to characterize the structure of Al_65_Cu_20_Fe_15_ alloy obtained with the use of conventional casting and rapid solidification-melt-spinning technology. Based on the literature data, the possibility of an icosahedral quasicrystalline phase forming in the Al-Cu-Fe was verified. Structure analysis was performed based on the results of X-ray diffraction, neutron diffraction, ^57^Fe Mössbauer and transmission electron microscopy. Studies using differential scanning calorimetry were carried out to describe the crystallization mechanism. Additionally, electrochemical tests were performed in order to characterize the influence of the structure and cooling rate on the corrosion resistance. On the basis of the structural studies, the formation of a metastable icosahedral phase and partial amorphous state of ribbon structure were demonstrated. The possibility of the formation of icosahedral quasicrystalline phase I-AlCuFe together with the crystalline phases was indicated by X-ray diffraction (XRD), neutron diffraction (ND) patterns, Mössbauer spectroscopy, high-resolution transmission electron microscopy (HRTEM) observations and differential scanning calorimetry (DSC) curves. The beneficial effect of the application of rapid solidification on the corrosive properties was also confirmed.

## 1. Introduction

The technologies of rapid solidification (RS), such as melt-spinning, water quenching, arc-melting, high pressure die casting, allow one to obtain a wide range of unique structures of aluminum alloys [1,2,3]. It is possible to fabricate Al-based alloys with structures such as: amorphous single phase, partially crystallized particles in an amorphous matrix, quasicrystals in Al matrix without grain boundary, and granular amorphous phases of aluminum with or without surrounding amorphous phase [2,4]. It seems particularly interesting to obtain quasicrystalline structures, which constitute the third state of a solid state next to crystalline and amorphous states [5]. The atoms in a quasicrystalline structure do not show the periodicity characteristic shown in those in a crystalline one. The quasicrystals demonstrate pentagonal, octagonal, decagonal, dodecagonal and icosahedral symmetries to a large extent, subject to a special quasiperiodicity principle [6]. Quasicrystalline structures are characterized by versatile properties, i.e., low thermal and electrical conductivity, low coefficient of friction along aperiodic axes, mechanical strength, high hardness, thermal stability and corrosion resistance [7,8]. Many alloying systems in which it is possible to obtain quasicrystalline phases were described in the literature, which proves their common occurrence. Metastable icosahedral phase systems are: Al-TM(V, Cr, Mn, Ru, Re) [9], Al-(Mn, Cr, Fe)-(Si, Ge) [9,10], Al-(Cu, Pd)-TM(Cr, Mn, Fe, Mo, Ru, Re, Os) [5,9,11]. Stable icosahedral phase systems include: Al_63_Cu_25_TM(Fe, Ru, Os)_12_ [9,12]. Metastable decagonal phase systems include: Al-TM (TM = Mn, Co, Fe, Pd) [9,13], Al-(Cu, Ni, Pd)-TM(Fe, Ru, Re, Co, Rh, Ir) [9]. Stable decagonal phase systems include: Al_70_Ni_15_Co_15_ [9], Al_65_Cu_15_Co_20_ [9,14], Al_75_Pd_15_TM(Fe, Ru, Os)_10_ [9]. In the literature, a lot of attention is paid to the ternary Al-Cu-Fe alloys due to their non-toxic, readily available, cheap and recycled chemical elements. The atomic fraction of Al-Cu-Fe alloys for which the icosahedral quasicrystalline structure can be obtained is given in the range: 23.4–25.1% Cu, 12.6–13.7% Fe, and 62.3–62.9% Al for conventionally casted alloys, 12.7–18.3% Cu, 14.7–19.6% Fe, and 63.3–68.6% Al for melt-spun ribbons, and 15.2–19.9% Cu, 14.2–18.6% Fe, and 62.3–68.3% Al for atomized powder [15]. The potential applications of quasicrystalline aluminum alloys with the addition of copper and iron include catalysts, elements of devices used in thermometry to detect heat flow, light absorbers in solar cells, element of bolometers in the detection of infrared radiation, and coating materials, for example in car engine pistons [16,17,18].

Based on the literature data, the possibility of a stable icosahedral quasicrystalline phase forming in the Al-Cu-Fe was verified [5,15]. The formation of a quasicrystalline structure in Al_65_Cu_20_Fe_15_ alloys was described in detail for in situ induction casted under an argon atmosphere, and also heat-treated [19], arc-melted [20,21], conventionally casted, melt-spun and created with the use of atomization techniques [15]. However, there is no work that compares the structures obtained in conventional casting and melt-spinning technologies with the use of a number of structural studies conducted by many various methods. Until now, there have been no data regarding the partial amorphous state of melt-spun Al_65_Cu_20_Fe_15_ alloy. Therefore, the aim of this work was to fill the research gap based on the results of structural studies and the mechanism of crystallization as well as corrosion resistance. The article was enriched with the results of electrochemical studies of ingot and melt-spun ribbon, which has also not been developed in other works. Structure analysis was performed based on the results of X-ray diffraction, neutron diffraction, and ^57^Fe Mössbauer in order to perform a phase composition analysis and confirm the occurrence of individual phases in all research techniques used. The transmission electron microscopy studies were carried out in order to characterize the structure at the atomic scale and confirm the phase composition. Studies using differential scanning calorimetry were also performed to describe the crystallization mechanism. Two different heating rates were used in order to accurately confirm the occurrence of thermal effects characteristic of phases transition.

## 2. Materials and Methods

Al_65_Cu_20_Fe_15_ alloys were fabricated using two different methods and cooling rates. The master alloys (ingots) were prepared by the induction melting of Al, Cu, and Fe (99.9%) under an argon atmosphere in a ceramic crucible. The samples in the form of ribbons were produced by the melt-spinning method by the Bühler Melt Spinner SC station (Edmund Bühler GmbH, Hechingen, Germany) with a copper wheel surface speed of 30 m/s using argon as protective atmosphere.

Phase identification was performed using X-ray diffraction (XRD) with a Cu Kα tube (λ = 0.154 nm). All samples were powdered and XRD patterns were recorded using a Mini Flex 600 equipped (Rigaku, Tokyo, Japan) with a copper tube as an X-ray radiation source and a D/TEX strip detector.

The neutron diffraction measurements were performed on the MTEST powder neutron diffractometer (Budapest Neutron Centre, Budapest, Hungary). The wavelength of the incident radiation was λ = 0.088 nm. The diffractometer was equipped with a Cu(220) monochromater, allowing the beam to be adjusted for the required Q-range and resolution.

To determine how the composition and casting method influenced the structure and Fe local environment, Mössbauer spectroscopy was applied as a complementary technique. ^57^Fe Mössbauer transmission spectra were recorded at room temperature with an MS96 Mössbauer spectrometer and a linear arrangement of a ^57^Co:Rh source, a multichannel analyzer with 1024 channels (before folding), an absorber, and a detector. The spectrometer was calibrated at room temperature with a 30 µm-thick α-Fe foil. Numerical analysis of the Mössbauer spectra was performed using the WMOSS program.

High-resolution transmission electron microscopy (HRTEM) was used to determine chemical composition, structure, and morphology using S/TEM TITAN 80–300 (FEI Company, Hillsboro, OR, USA). The EDS analysis in the high-angle annular dark-field mode (HAADF-STEM) was used to confirm a composition of tested samples. Samples for HRTEM observations were prepared by FIB method.

To determine the crystallization mechanism, differential scanning calorimetry (DSC) of all studied samples was performed using thermal analyzer SDT Q600 (TA Instruments, New Castle, DE, USA). The DSC curves were recorded at heating rates of 10 and 20 °C/min and a cooling rate of 10 °C/min under a protective argon atmosphere.

In order to compare the two different cooling rates used for analyzed alloys, electrochemical measurements in a 3.5% NaCl solution using an Autolab 302N were performed. The potentiostat (Metrohm AG, Herisau, Switzerland) was equipped with a three-electrode cell controlled by NOVA software. The reference electrode was saturated calomel electrode (SCE) and the counter electrode was a platinum rod. The corrosion resistance was measured by the open-circuit potential (E_OCP_) variation in the function of the SCE. Samples were measured after 3600 s of open-circuit potential stabilization at a scan rate of 1 mV s^−1^.

## 3. Results

### 3.1. Structure Analysis

Figure 1 shows the X-ray diffraction patterns of the Al_65_Cu_20_Fe_15_ alloy in as-cast ingot and melt-spun ribbon states. Main diffraction peaks of the ingot alloy can be identified as the AlCuFe icosahedral phase [18], and the other diffraction peaks correspond to multiple phases of Al_2_Cu, Al_7_Cu_2_Fe, AlFe, Al_13_Fe_4_ and Cu_3_Al (Figure 1a). In other works, the following phases were obtained for as-cast alloys: I-AlCuFe, Al_13_Fe_4_ and Al(Cu,Fe) [21], I-AlCuFe, Al_13_Fe_4_, AlFe, Al_2_Cu [22], I-AlCuFe, Al_13_Fe_4_ with increasing content disordered B2 phase after mechanical milling [23] and I-phase, Al_13_Fe_4_ and Al(Cu,Fe) [15]. In other works, the Al_65_Cu_20_Fe_15_ alloy used as a galvanic coating was characterized by the phase composition: quasicrystalline phases, Al_65_Cu_20_Fe_15_, Al_2_Cu, and AlCu_3_ [24]. The melt spinning involved the formation of an amorphous phase, and the icosahedral phase is a metastable phase caused by rapid solidification. The presence of Al, Al_2_Cu, and Al_13_Fe_4_ was clearly detected (Figure 1b). The broadening of diffraction lines (Figure 1a) implies that the melt-spun ribbon is partly amorphous. The presence of phases was also identified for ribbons in other articles: I-phase, Al_13_Fe_4_ and Al(Cu,Fe) [15] and I-phase, AlFe [22]. Due to the limitation of the resolution of the conventional XRD method, neutron diffraction (ND) measurements were provided for the melt-spun ribbon. Figure 2 presents the ND pattern with identified phases. The spectrum confirms the presence of the I-phase and Al_13_Fe_4_, Al_2_Cu, and Al, which were detected by XRD patterns.

To clarify the detailed structure of studied alloy, the HRTEM examinations were carried out and two different regions were observed in the melt-spun sample. Figure 3 shows a bright-field electron micrograph (Figure 3a) and selected-area electron diffraction patterns (Figure 3b) of the first region with the homogenous morphology. Figure 3g shows a representative bright-field TEM micrograph showing the microstructure of the second region. The image showed similar microstructure containing crystalline particles. To further investigate the detailed structure of melt-spun ribbon, the high-resolution image was chosen for Fourier transfer (FT) and inverse Fourier transfer (IFT) treatments. Figure 3c–f present the IFT images from the selected areas 1–4. Similarly, for the second region, the four regions (5–8) marked in Figure 3g, the IFT pictures of which are shown in Figure 3i–l, are specified. It can be noticed that some parts from the disorder structure can be observed in the vicinity of quasicrystals. The relationship between the formation of three types of structures—quasicrystalline, amorphous, and crystalline—in the Al_65_Cu_20_Fe_15_ alloy were observed by Chien and Lu when obtaining layers by sputtering and annealing at various temperatures [25]. The areas with specific arrangements of atoms (labeled by the circles) indicate that the icosahedral short-range order can possibly exist in the studied alloy. The ordering of short range quasicrystals in the Al_65_Cu_20_Fe_15_ alloy was also indicated by Sadoc, pointing out that Fe and Cu atoms do not occupy the same sites at random [26]. The IFT images also show areas with an ordered structure of atoms, characteristic of crystalline structures, which is consistent with the literature [27], where the possibility of the coexistence of crystalline, quasicrystalline, and amorphous phases was confirmed under appropriate conditions for the production of Al-Cu-Fe alloys. Selected area electron diffraction (SAED), presented in Figure 3b,h, confirmed the presence of Al, Al_13_Fe_4_, and Al_2_Cu phases in the studied ribbon. The SAED patterns indicate that the structure was dominated by the Al_2_Cu phase which, together with the Al_13_Fe_4_ phase, indicated numerous peaks in XRD patterns. Discrepancies in the XRD and TEM phase analysis result from the size of the analyzed areas, which was also indicated in the work of Shadangi et al. [28]. In addition, for Al_65_Cu_20_Fe_15_ alloys in other studies, the presence of crystalline phases in the predominance of crystalline phases was similarly indicated in other studies: for powder milled-Al(Fe, Cu) [29]. Additionally, HAADF images and EDS spectra were obtained from local areas of the melt-spun sample and confirmed the qualitative chemical composition of studied alloy, as shown in Figure 4.

The ^57^Fe Mössbauer spectroscopy method (as a local probe technique) is very sensitive to the local atomic structure, its local deformation, and atomic or lattice defects such as vacancies or dislocations when treating the Fe nucleus as a probe of its local surrounding. The obtained ^57^Fe Mössbauer spectra with the fitted components for the investigated Al_65_Cu_20_Fe_15_ alloy in an as-cast state and in the form of ribbon are presented in Figure 5. The hyperfine interaction parameters obtained from the analysis of the spectra are summarized in Table 1. The Mössbauer spectrum of the alloy in the form of ribbon can be fitted by the three singlets corresponding to the Al_13_Fe_4_ phase [30,31] and also by two doublets. The first doublet (Is = 0.19 mm/s, Qs = 0.73 mm/s) is associated with the Al phase rich in Cu and Fe elements. The second doublet will be related to the quasicrystalline Al-Cu-Fe phase.

The Mössbauer spectrum of the master alloy (ingot) contains four singlets and two quadrupole doublets. The values of the isomer shift of the first three singlets are typical for the Al_13_Fe_4_ phase and the fourth singlet (Is = 0.14 mm/s) is connected with the Al_7_Cu_2_Fe phase [31]. The doublet with Is = 0.34 mm/s and Qs = 0.24 mm/s is related to iron located in body centered cubic structure of AlFe(Cu) [31]. According to the XRD results, the doublet with the highest contribution was associated with the quasicrystalline Al-Cu-Fe phase.

The crystallization mechanism of the studied alloys was proposed based on the DSC curves in Figure 6 with heating and cooling rates of 10 °C/min and Figure 7 with a heating rate of 20 °C/min and a cooling rate of 10 °C/min for ingots (a) and for ribbons (b). The DSC curves for ribbons are presented in the temperature range of 700–1030 °C, while for ingots, measurements for the heating rate of 20 °C/min are shown in the range of 700–1075 °C. In order to indicate the identified Al_2_Cu phase dissolution, the measurements for 10 °C/min are illustrated in a range of 400–1050 °C. Additionally, the phases assigned to the temperatures of thermal effects data are summarized in Table 2. DSC heating curves of an ingot showed three endothermic peaks for 10 °C/min and four endothermic peaks for 20 °C/min, and the form of a ribbon indicated two endothermic peaks for 10 and 20 °C/min. A slight endothermic effect for alloy in the form of an ingot at the temperature of 643 °C for the rate of 10 °C/min is associated with the dissolution of the Al_2_Cu phase, which is characterized by a low enthalpy for the Al_65_Cu_20_Fe_15_ alloys [5]. The endothermic effects around 863 (10 °C/min) and 843 °C (20 °C/min) for the ingot as well as 863 (10 °C/min) and 862 °C (20 °C/min) for the ribbon correspond with the decomposition of I-AlCuFe. Literature data indicate that the temperature of the decomposition of the icosahedral phase of AlCuFe is 880–890 °C [5,22]. However, according to the Al-Cu-Fe phase diagram presented in the literature [32], the melting point of the phases could be shifted, which is related to the local differences in stoichiometry [5]. Similarly, for the AlFe phase, the melting point was reported to be around 1000 °C. In this work, the temperature for the AlFe phase is characterized at around 920 °C for ingots, with a 20 °C/min cooling rate. Based on the literature data, the effects around 977 (ingot, 20 °C/min), 982 (ingot, 20 °C/min) and 975 °C (ribbon, 10 °C/min) could indicate the melting of intermetallic phase Al_13_Fe_4_ (around 1000 °C) [33]. Similarly, the temperatures of 1010 (ingot, 10 °/min) and 1039 °C (ingot, 20 °/min) correspond to the Cu_3_Al phase (around 1050 °C) [34]. In the case of the 20 °C/min heating curve for the ribbon, the second endothermic peak is markedly widened, with no distinct onset of transformation, possibly indicating an overlapping of the melting transformations for the Al_13_Fe_4_ and Cu_3_Al phases. The crystallization of ingots for the 10 °C/min cooling rate indicates the presence of three distinct endothermic peaks. The precipitation of the solid phase was started around 1024 °C. The first peak indicates crystallization of the Al_3_Cu phase. Then, the superimposition of thermal effects related to the crystallization of Al_13_Fe_4_ and AlFe phases is visible. The effects around 1055, 1004, 895, and 801 °C correspond to crystallization of phases: Cu_3_Al, Al_13_Fe_4_, AlFe, I-AlCuFe, respectively, for ingots with a cooling rate of 20 °C/min. In the case of a ribbon cooled at a rate of 10 °C/min, three peaks are visible, which are assigned to phase crystallization: Cu_3_Al, Al_13_Fe_4_, I-AlCuFe. The cooling curve for 20 °C/min also indicated three effects of 1016, 957, and 831 °C connected with the formation of the Cu_3_Al, Al_13_Fe_4_ and I-AlCuFe phases, which is in accordance with the volume of these phases in equilibrium in analyzed temperature range [5].

### 3.2. Corrosion Behavior

The electrochemical studies were performed to analyze the influence of structural changes on corrosion resistance between two different cooling rates. Figure 8 presents the open-circuit potential plots (a) and polarization curves (b) of the studied samples. The results of open-circuit potential (E_OCP_), corrosion potential (E_corr_), anodic and cathodic Tafel’s slopes (β_a_, β_c_), polarization resistance (R_p_), corrosion current density (j_corr_) for the ingot and ribbon are listed in Table 3. On the basis of the presented results, higher values of the open circuit potential for the ribbon can be observed in Figure 8a and in Table 3. Likewise, the favorable values for the ribbon are visible in the case of a shift of the polarization curve towards more positive E_corr_ values and lower corrosion current density values, which is also confirmed by the electrochemical data. The comparison of the values of open circuit potential and corrosion potential is used in assessing corrosion resistance between individual studied samples under the same conditions. This approach was also described in the literature [35,36]. In the case of using higher cooling rate, the value of the polarization resistance was much higher compared to the slow-cooled alloy, which in combination with the other values indicates an improvement in the anti-corrosion properties. The improvement of the corrosion resistance of alloys with a higher cooling rate has been described in the literature [37,38]. This is due to the fragmentation of the structure which is visible by the broadening of the diffraction peaks in the XRD pattern (Figure 1).

## 4. Conclusions

The article presents an analysis of the results of structural studies and the crystallization mechanism of the Al_65_Cu_20_Fe_15_ alloy prepared by two different cooling rates from the liquid state: conventional casting and the rapid solidification-melt-spinning method. In the detailed work, a number of results of structural studies were developed using various advanced methods, which supplements the actual knowledge. The possibility of the formation of an icosahedral quasicrystalline phase I-AlCuFe together with the crystalline phases was indicated by XRD, ND patterns, Mössbauer spectroscopy, HRTEM observations and DSC curves. In the case of the ribbon, a broadening of the diffraction lines was obtained, indicating a partial amorphous state. The tendency towards amorphization of the studied alloy was not described so far for the melt-spinning technology. In this work, it was indicated by X-ray diffraction method and HRTEM observations, which a new knowledge. Based on the DSC heating and cooling measurements, a crystallization mechanism was described which differed depending on the heating and cooling rate used. However, for all measurements, there were thermal effects characteristic of phase I-AlCuFe transformations at the temperature around 862–863 °C (heating). The remaining endothermic peaks characteristic for the other identified phases differed due to local differences in stoichiometry of the examined samples. Additionally, electrochemical measurements were performed to determine the effect of rapid cooling on the corrosion resistance of Al_65_Cu_20_Fe_15_ alloy. It was shown that the ribbon exhibited more favorable values for open-circuit potential (−665 mV for the ingot, −657 mV for the ribbon), corrosion potential (−482 mV for the ingot, −396 mV for the ribbon), polarization resistance (2.23 kΩ∙cm^2^ for the ingot, 8.74 kΩ∙cm^2^ for the ribbon) and corrosion current density (3.59 µA/cm^2^ for the ingot, 2.42 µA/cm^2^ for the ribbon). The obtained results for melt-spun Al_65_Cu_20_Fe_15_ alloys could be the basis for further studies on the formation of amorphous-quasicrystalline corrosion-resistant alloys.

## Figures and Tables

**Figure 1 materials-14-00054-f001:**
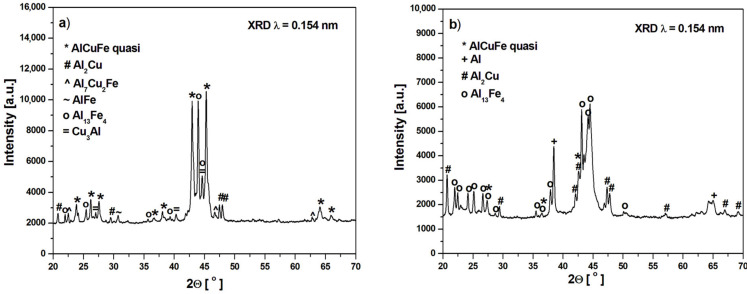
XRD pattern of Al_65_Cu_20_Fe_15_ alloy in the form of ingot (**a**) and ribbon (**b**).

**Figure 2 materials-14-00054-f002:**
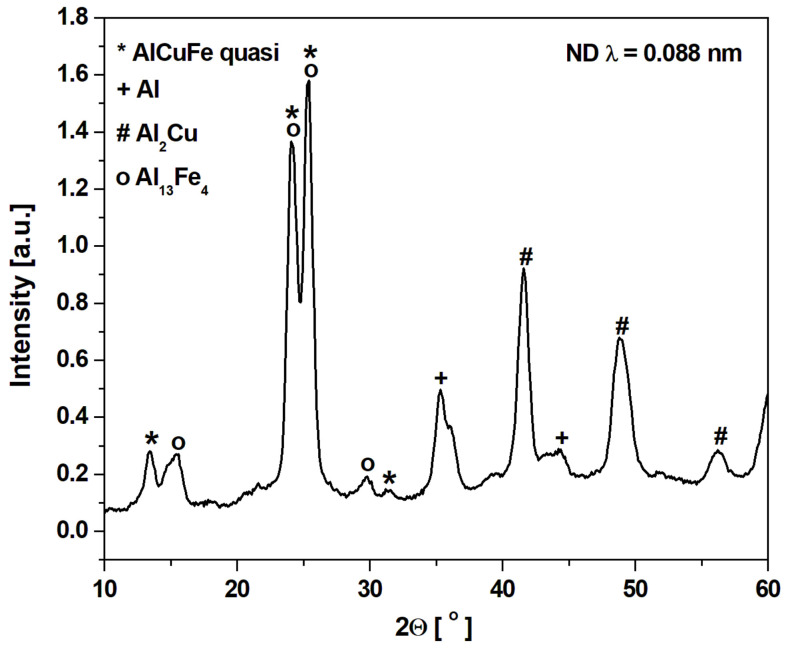
Neutron diffraction (ND) pattern of Al_65_Cu_20_Fe_15_ alloy in the form of a ribbon.

**Figure 3 materials-14-00054-f003:**
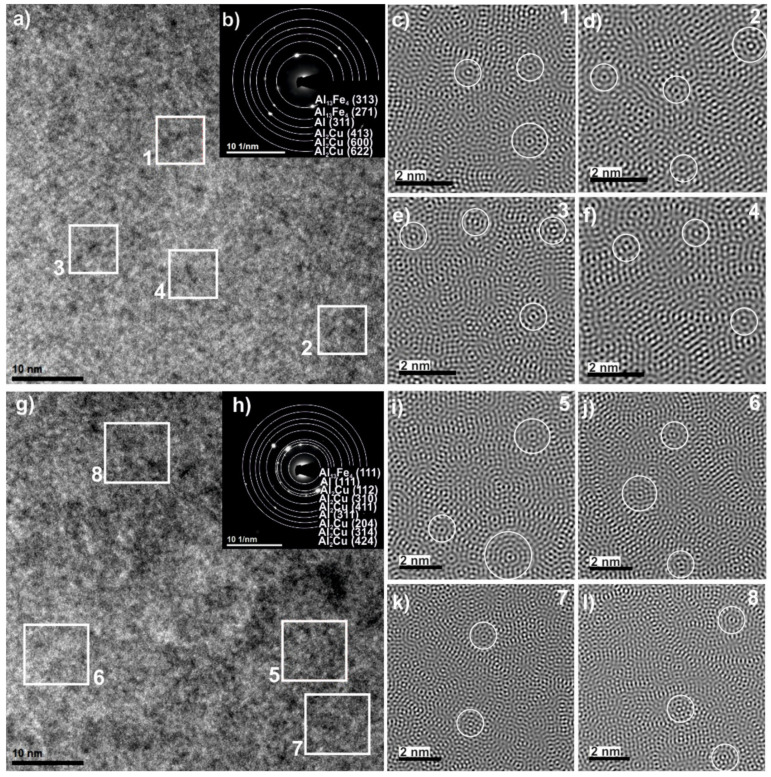
High-resolution transmission electron microscopy (HRTEM) images (**a**,**g**), selected area electron diffraction (SAED) patterns (**b**,**h**) of Al_65_Cu_20_Fe_15_ alloy in the form of ribbon and inverse Fourier transfer (IFT) images (**c**–**f**,**i**–**l**) from selected areas 1–8.

**Figure 4 materials-14-00054-f004:**
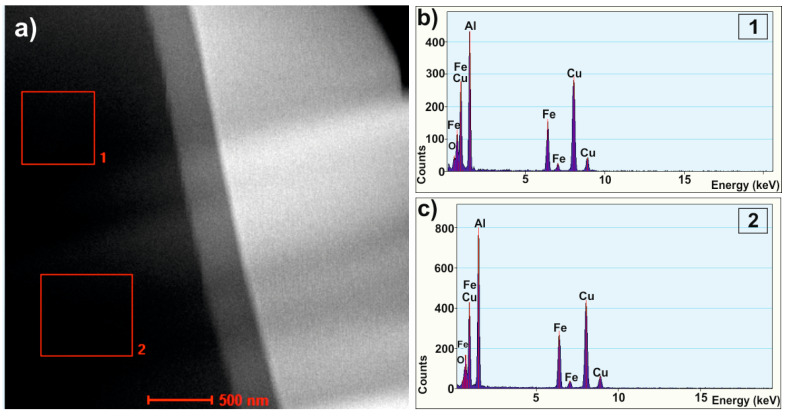
HAADF image (**a**) and EDS spectra (**b**,**c**) from selected areas (**1**,**2**) of Al_65_Cu_20_Fe_15_ alloy in the form of ribbon.

**Figure 5 materials-14-00054-f005:**
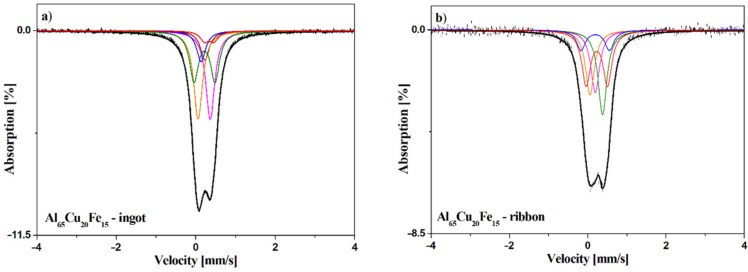
Mössbauer spectra for the alloys recorded for Al_65_Cu_20_Fe_15_ ingot (**a**) and ribbon (**b**).

**Figure 6 materials-14-00054-f006:**
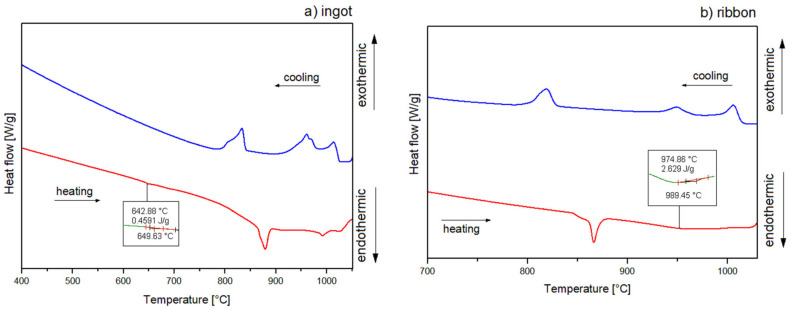
Differential scanning calorimetry (DSC) curves of Al_65_Cu_20_Fe_15_ alloy for (**a**) ingot, (**b**) ribbon, 10 °/min.

**Figure 7 materials-14-00054-f007:**
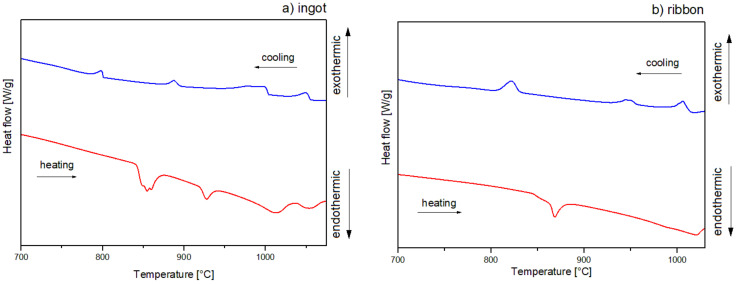
DSC curves of Al_65_Cu_20_Fe_15_ alloy for (**a**) ingot, (**b**) ribbon, 20 °/min.

**Figure 8 materials-14-00054-f008:**
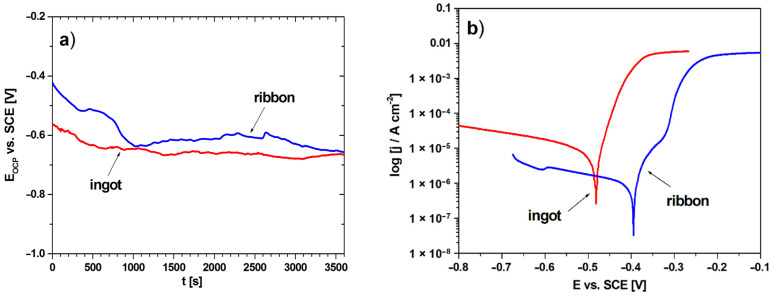
Variation in the open-circuit potential with time (**a**) and polarization curves (**b**) for ingot and ribbon in 3.5% NaCl solution at 25 °C.

**Table 1 materials-14-00054-t001:** The Mössbauer hyperfine parameters of the investigated samples. Isomer shift (Is), quadrupole splitting (Qs), full width at half maximum (FWHM), and relative area from the spectra (A) for the studied samples.

Component	Is [mm/s]	Qs [mm/s]	FWHM [mm/s]	A [%]	Compound
**Al_65_Cu_20_Fe_15_—Ingot**					
L1	0.05	-	0.27	24	Al_13_Fe_4_
L2	0.23	-		9	
L3	0.36	-		24	
L4	0.14	-		10	Al_7_Cu_2_Fe
D1	0.34	0.24		5	AlFe
D2	0.21	0.53		29	AlCuFe quasi
**Al_65_Cu_20_Fe_15_—Ribbon**					
L1	0.06	-	0.27	18	Al_13_Fe_4_
L2	0.19	-		18	
L3	0.38	-		23	
D1	0.19	0.73		12	Al(Fe,Cu)
D2	0.23	0.54		29	AlCuFe quasi

**Table 2 materials-14-00054-t002:** Phases identified based on XRD patterns with DSC data.

Alloy		Identified Phases	Heating 10 °/min [°C]	Cooling 10 °/min [°C]	Heating 20 °/min [°C]	Cooling 20 °/min [°C]
Al_65_Cu_20_Fe_15_	ingot	Al_2_Cu	643	-	-	-
		I-AlCuFe	863	840	843	801
		AlFe	-	-	920	895
		Al_13_Fe_4_	982	1024	977	1004
		Cu_3_Al	1010	1024	1039	1055
	ribbon	I-AlCuFe	862	823	863	831
		Al_13_Fe_4_	975	963	-	957
		Cu_3_Al	-	1014	-	1016

**Table 3 materials-14-00054-t003:** The polarization tests of Al_65_Cu_20_Fe_15_ alloy in the form of an ingot and ribbon in 3.5% NaCl solution at 25 °C (E_OCP_—open-circuit potential, E_corr_—corrosion potential, β_a_, β_c_—anodic and cathodic Tafel’s slopes, R_p_—polarization resistance, j_corr_—corrosion current density).

Sample	E_OCP_ [mV]	E_corr_ [mV]	|β_a_| [mV/dec]	|β_c_| [mV/dec]	R_p_ [kΩ∙cm^2^]	j_corr_ [µA/cm^2^]
ingot	−665	−482	90.8	23.2	2.23	3.59
ribbon	−657	−396	86.1	33.8	8.74	2.42

## Data Availability

Data sharing is not applicable to this article.

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
