# Peer review of "Structural Characterization of Al65Cu20Fe15 Melt-Spun Alloy by X-ray, Neutron Diffraction, High-Resolution Electron Microscopy and Mössbauer Spectroscopy"

_materials, 2020, doi:10.3390/ma14010054_

Round 1
Reviewer 1 Report
There are many publications on the Structural characterization of Al65Cu20Fe15 available online, such as
1. Characterisation of Al65Cu20Fe15 Quasicrystal Alloy Synthesised via In-situ Casting under Standard Room Ambient and Argon Enriched Atmosphere
- March 2018
- International Journal of Current Research in Science Engineering & Technology 1(Spl-1):280
- DOI: 10.30967/ijcrset.1.S1.2018.280-286
2. Characterization of the icosahedral phase in as-cast quasicrystalline Al65CU20Fe15 alloy
- September 2001
- Materials Characterization 47(3-4):299-305
- DOI: 10.1016/S1044-5803(02)00182-1
3. A study on spherical particles in Al65Cu20Fe15 alloy prepared by arc melting
4.Characterization of Rapidly Solidied Al65Cu20Fe15 Alloy in Form of Powder or Ribbon
- DOI: 10.12693/APhysPolA.126.512
and other publications, can the author provide what has been done differently in the manuscript compare to what is already published?
Why not use the same scale in Figure 8?
Author Response
Please see the attachment.
Kind regards
Katarzyna Młynarek

Reviewer 2 Report
I think this is an excellent study. I suggest the study can be published after minor revisions.
- The last paragraph (hypothesis section) should be more quantified in the introduction. Authors can mention what testing they performed and why they performed it rather than mentioning it in the individual section. That will be better for understanding what to expect from a reader's perspective.
- The introduction mentions conventional casting and rapid solidification-melt-spinning technology but fails to shed light on any previous studies on this topic. If so, then how the current study differ from that?
- Figure 3 and Figure 4 are basically the same but from two different areas. Initially, it was misleading. But I would suggest putting either one of them to supplement or merge them in one figure with a subsection.
- The author has chosen EDS but never mention the atomic percentages they observed in EDS. What are the significant changes they observed? I think they can put it in to supplement.
- In conclusion, the authors need to put electrochemical corrosion measurements more quantifiable, such as what values were achieved in ingot and ribbon in corrosion potential, polarization resistance, etc., melt spinning more favorable compared to casting.
Author Response

(The authors gave the same response as above.)

Reviewer 3 Report
The most important problem is that there is no description on what is not known or not clarifed in this field in the "introduction". Therefore, the readers can not know what the authors try to explore in the study. Conclusions is a collection of experimental facts, there is no description of what you want to "conclude". Thus, we cannot understand the significance or contrictribution of the paper to this field. There are several simple mistakes, for example, in the first three lines of introductions, "water-quenching, ---water quenching, high pressure, ---- high pressure" etc. The reviewers are discouraged to read the paper furthrer, since it was not prepared with careful check.
Author Response

(The authors gave the same response as above.)

Reviewer 4 Report
See comments submitted separately.

Author Response

(The authors gave the same response as above.)
